# Overestimation of Phonological Judgments on the Right Side of Space

**DOI:** 10.3390/brainsci13081123

**Published:** 2023-07-25

**Authors:** Patrizia Turriziani, Alessia Santostefano, Angela Catania, Massimiliano Oliveri

**Affiliations:** 1Department of Psychology, Educational Science and Human Movement, University of Palermo, 90128 Palermo, Italy; 2NeuroTeam Life and Science, 90143 Palermo, Italy; alessia.santostefano3@gmail.com (A.S.); 3International School of Advanced Studies, University of Camerino, 62032 Camerino, Italy; angela.catania@unicam.it (A.C.); 4Biomedicine, Neurosciences and Advanced Diagnostics Department, University of Palermo, 90128 Palermo, Italy; massimiliano.oliveri@unipa.it (M.O.)

**Keywords:** attention, phonological processing, space

## Abstract

Spatial attentional biases can be observed during the processing of linguistic material. For example, we previously reported that healthy subjects overestimate the semantic distance between word stimuli in the right vs. left space. Here, we explored whether or not attentional biases are also observed in tasks requiring an evaluation of phonological distance between words in the right and left hemifield. Forty-one healthy subjects were presented with triplets of words arranged in space and were asked to indicate the side of the space in which the phonological distance between the middle word and an outer word was smaller. In Experiment 1, real words and pseudowords were used, while in Experiment 2, only pseudowords and consonant strings were used. Subjects overestimated the phonological distance between the middle and outer words in the right space. These findings were specific to word stimuli. These results are consistent with the idea that semantic and phonological information may be internally mapped onto spatial representations.

## 1. Introduction

The distribution of attention and mental representation of a space seem asymmetrical, as demonstrated by biases displayed by healthy subjects in attentional tasks using a wide range of stimuli.

The most well-known form of attentional bias is pseudoneglect, initially described as a leftward bias in the bisection of physical lines [1,2,3]. Indeed, pseudoneglect is observed in different domains. In number comparison tasks, healthy subjects overestimate the difference between a middle number and an outer number on its left side [3,4]. Healthy participants show a leftward bias even when they are demonstrated three-letter strings and are asked to estimate which of the two flankers (e.g., C and P) has a greater alphabetical distance from the inner letter (H) [5,6].

It remains to be seen whether there may also be attentional biases for linguistic domains without explicit left-to-right representation. Studies investigating the processing of letters that need to be organised alphabetically have yielded conflicting results. In letter line bisection tasks, some studies documented a bias towards the left hemispace [7], whilst others documented a bias towards the right hemispace [8].

Mohr and Leonard [9] investigated the impact of semantic information on letter line bisection in healthy subjects. The authors used letter lines with embedded words that were either emotional (e.g., eucsoiaad*kill*fp) or neutral (e.g., heaiineb*main*ul). A stronger rightward bisection bias for letter lines containing emotional words was documented. The authors suggested that the semantic information activated the left hemisphere more strongly than it did the right hemisphere, resulting in a rightward shift of processing. Turriziani and colleagues [10] explored whether or not semantic judgments could be modulated by the location in a space where a stimulus conveying semantic information was presented. Healthy subjects viewed three pictures of items in the same semantic category arranged horizontally, one on the left side, one in the middle, and one on the right side. On average, the semantic distance between the middle picture and left one was reported as smaller than the semantic distance between the middle picture and the right one. These findings suggest the existence of an attentional and mental representational bias in semantic judgements, like those reported for spaces, numbers and alphabet lines. In addition, rTMS over the left parietal cortex selectively reduced this rightward bias. This suggests that spatial manipulation of semantic material could result in the activation of specialised attentional resources located in the left hemisphere.

Motivated by the wealth of previous research, the present study aims to further explore the biases previously reported in semantic judgment tasks and investigate whether these biases are specific to semantic representation or can be extended to other components of language, such as phonology. By examining the influence of spatial location on phonological tasks requiring participants to judge the side of space where the phonological distance between a middle word and two outer words is smaller, we aim to uncover the intricate interplay between attention, spatial cognition, and language processing. This investigation has the potential to provide valuable insights into the underlying mechanisms of attentional biases in linguistic domains and expand our understanding of the complexities of human cognition.

## 2. Experiment 1

This experiment investigated the relationship between the phonological distance among words and the space in which these words were presented.

### 2.1. Materials and Methods

#### 2.1.1. Subjects

Sixteen right-handed native Italian speakers (6 M and 10 F; mean age: 25 ± 2.9 years) with normal or corrected-to-normal vision and naïve to the purpose of the study were enrolled. All subjects gave written informed consent for participation in the study which was approved by the ethical committee of the University of Palermo (approval n. 25/2020). The experiments were conducted in accordance with the principles of the Declaration of Helsinki.

#### 2.1.2. Materials

Briefly, 280 four- or five-letter words and 40 five-letter pronounceable pseudowords were selected. These 320 words were combined to obtain 120 different triplets. Each triplet comprised a middle and two outer words. Triplets were presented for 550 ms on a 19-inch 50 Hz computer monitor. The middle word was presented in the centre of the monitor. The two outer words were presented with 5° of eccentricity to the left and right of the middle word (the gap between the middle and either outer word was 5°; Figure 1a). The intertrial interval was 2500 ms. Participants were seated 45 cm from the monitor and were asked to focus on a central fixation cross that preceded the item’s presentation. The phonological distance between the middle and the two outer words could be the same or smaller on the right or left side of space.

There were three experimental conditions: Same, Different and Very Different conditions. In the Same condition, triplets were composed of three words that differed only in the first letter (e.g., *mela*, *vela*, and *gela*) and with an identical phonological distance (i.e., with the same phonological “sound” of the whole word) between the middle and the two outer words. In the Different conditions (e.g., *pollo*, *collo*, and *cesta*), one outer word of the triplet was phonologically close to the middle one. In contrast, the other outer word was similar to the middle word only for the first letter (e.g., *collo*, and *cesta*). Finally, in Very Different conditions, the triplets comprised two words and one pseudoword. Therefore, the phonological distance between the middle word and one of the two outer words was much smaller than that between the middle word and the pseudoword (e.g., *gatto*, *matto*, and *fupro*).

There were 40 triplets in each of the three experimental conditions.

#### 2.1.3. Procedure

All subjects received training in the testing procedure, until they felt confident to start the experiment. In the training, they were presented with word triplets that were different from those included in the main experiment and familiarised with the experimental question of “phonological distance” referring to the sound of the word when pronounced. Subjects were asked to indicate the side of the space in which the phonological distance between the middle word and an outer word was smaller (“Where is the word phonologically closest to the middle word?”). Participants were told to choose the “same” response if neither of the two outer words appeared more phonologically related to the middle item. Participants responded by pressing one of three buttons with the right middle, index or ring finger for the “same”, “left”, or “right” responses, respectively. The side of space in which the target words appeared within each triplet was randomised.

### 2.2. Results

Accuracy (mean number of errors) and reaction times for correct responses (RTs: interval of time between the onset of stimuli and the participant’s response) were analysed.

The participants’ responses in each experimental condition are shown in Table 1.

We performed a 3 × 2 ANOVA on the mean number of errors, with the variables Condition (Different, Same, and Very Different) and Space (left, right) as within-subjects factors. As shown in Figure 1b, there was a significant main effect of Condition (F_2,30_ = 13.67, *p* < 0.001). This reflects the fact that the average number of errors in the Very Different condition was significantly different from both the Same (F_1,15_ = 35.23, *p* < 0.001) and Different (F_1,15_ = 16.99, *p* < 0.001) conditions. The error rates in the Same and Different conditions were comparable (F_2,18_ = 3.7, *p* > 0.5). The main effect of Space was not significant (F_1,15_ = 0.001, *p* > 0.5). The interaction of Condition x Space was significant (F_2,30_ = 11.4, *p* < 0.005). Planned comparisons revealed a rightward bias in the Different conditions. Specifically, in trials where the phonological distance was smaller between the middle and the outer word positioned in the right space, participants tended to produce erroneous “left” or “same” responses (F_1,15_ = 11.19, *p* < 0.005). This means that subjects made two type of errors in this condition: (1) they erroneously chose the left instead of the right outer word as the one with the shorter phonological distance from the middle one; (2) they erroneously considered the two outer words as having the same phonological distance from the middle one. In both cases, this pattern of responses reflects an overestimation of the phonological distance in the right space, an overestimation that is even greater in case 1, when subjects shifted the shorter judgment from the right to the left hemifield.

In the Same condition, participants erroneously judged the phonological distance between the middle and the outer word positioned in the left space to be smaller, at F_1,15_ = 4.67, *p* < 0.05). The greater number of erroneous judgments of the phonological distance as shorter in the left hemifield in the Same condition reflects a shift towards the overestimation of phonological distance in the right hemifield. There was no significant difference between leftward and rightward biases in the Very Different condition (F_1,15_ = 2.3, *p* < 0.5).

The ANOVA performed on the RTs did not reveal differences between the Same and Different conditions with target outer words positioned in the left or right space (F_2,30_ = 0.85, *p* > 0.5). In the Very Different condition, there were no significant differences between triplets with target outer words in the left and right space (F_2,30_ = 0.29, *p* > 0.5).

In sum, phonological judgements were influenced by the spatial location of the stimuli. For example, when comparing the phonological distance between pairs of stimuli, subjects tended to overestimate the distance between a middle word and an outer word positioned to its right.

## 3. Experiment 2

This experiment examined whether or not the direction of attentional bias demonstrated in Experiment 1 is also observed in tasks involving phonological judgements of words without semantic representation, such as pronounceable pseudowords.

### 3.1. Materials and Methods

#### 3.1.1. Subjects

Twenty-five right-handed subjects (7 M and 19 F; mean age: 24 ± 3.1 years) participated in this experiment. None of them participated in Experiment 1.

#### 3.1.2. Materials

The experimental procedure was identical to that of Experiment 1, except for the stimuli used.

Briefly, 280 four- or five-letter pseudowords and 40 consonant strings on five letters were selected. These 320 stimuli were combined to obtain 120 different triplets. Each triplet was constituted by a middle and two outer pseudowords. The two outer pseudowords were presented with 5° of eccentricity to the left and right of the middle pseudoword (Figure 2a).

There were three experimental conditions: the Same, Different and Very Different conditions. In the Same condition, triplets were composed of three pseudowords with an identical phonological distance between the middle and the two outer pseudowords (e.g., *dali*, *fali*, and *rali*). In the Different condition, the triplet was composed of three pseudowords, and the phonological distance between the middle and one of the two outer pseudowords was smaller (e.g., *tresa*, *tarto*, and *marto*). The nearer phonological distance was when the middle and one outer pseudoword differed only in the first letter (e.g., *tarto* and *marto*). Likewise, the other outer word could be similar to the middle word only for the first letter (e.g., *tresa*, and *tarto*). In the Very Different condition, the triplets were composed of two pseudowords and one consonant string, and the phonological distance between the middle and one of the two outer pseudowords was smaller than the phonological distance with the other outer pseudoword (e.g., *gpnt*, *pito*, and *nito*).

### 3.2. Results

We performed a 3 × 2 ANOVA on the mean number of errors, with the variables Condition (Different, Same, and Very Different) and Space (left and right) as within-subject factors. As shown in Figure 2b, there was a significant main effect of Condition (F_2,48_ = 3.00, *p* = 0.05), with the lowest error rates observed for the Very Different condition. Furthermore, the average number of errors in the Very Different condition was significantly different from that in both the Same (F_1,24_ = 4.89, *p* < 0.05) and Different (F_1,24_ = 5.58, *p* < 0.05) conditions. However, the error rates in the Same and Different conditions were comparable (F_1,24_ = 0.04, *p* > 0.5).

The main effect of Space was not significant (F_1,24_ = 0.00, *p* > 0.5). Also, the interaction of Condition × Space was not significant (F_2,24_ = 2.14, *p* > 0.5).

The ANOVA performed on the RT data did not reveal differences between the Same and Different conditions with target outer words positioned in the left or right space (F_2,48_ = 0.63, *p* > 0.5). In the Very Different condition, there were no significant differences between triplets with target outer words in the left and right space (F_2,48_ = 0.18, *p* > 0.5).

This experiment’s results show that the stimuli’s spatial location did not influence the phonological judgments of pseudowords.

## 4. Discussion

The present study provides valuable insights into the interaction between spatial attention and linguistic processing, specifically in the domain of phonology.

The main results of the present study show that when asked to compare the phonological distances between words, subjects consistently overestimate phonological distances on the right side of space. This overestimation can be the consequence of two type of errors: 1. erroneous judgment of the phonological distance between the outer left word and the middle word as shorter in the trials where the phonological distance was in fact shorter between the outer right word and the middle one (“Different” trials); 2. erroneous judgment of the phonological distance between the outer left word and the middle word as shorter in the trials where the phonological distance was in fact the same between the outer right word and the middle one (“Same” trials).

Semantic information in phonological strings is crucial in determining this rightward bias. Indeed, healthy subjects show a rightward bias in a bisection task involving phonological distances when the stimuli are real words (Experiment 1) but not when the stimuli are pseudowords (Experiment 2).

The finding that the spatial location of verbal stimuli modulates the performance on a phonological task is in line with the results of other studies, reporting that spatial manipulation of semantic information induces a rightward bias in a semantic judgement task [10]. These findings suggest that real language, conveying semantic, phonological and syntactic information, may be internally mapped onto spatial representations.

Interestingly, the type of stimuli processed influences the interaction between space and phonology. For example, a significant rightward bias in the phonological distance judgment task was present only when real words were presented. On the other hand, the mere presentation of verbal material, for example, pseudowords, did not induce any spatial attentional bias. Therefore, the semantic component of language interacts with spatial attention, either when processed explicitly, as in Turriziani et al. [10], or implicitly, as in the present study.

These results show for the first time that phonology is a language component that can be processed with reference to spatial components.

Evidence from neuropsychological studies is also suggesting a link between space and other language components. Coslett reported that in some aphasics, the direction in which they orient their attention influences their use of language [11]. Chatterjee and colleagues [12] described an agrammatic patient whose production and comprehension of sentences were influenced by spatial factors. Rinaldi and Pizzamiglio [13] reported that patients with left spatial neglect made significantly more errors when asked to compare two spoken sentences if the emphatic stress was placed at the beginning of the sentence. Overall, these results seem to support Coslett’s “Spatial Registration Hypothesis” [11], suggesting that each perceived stimulus is automatically marked with reference to its coordinates in egocentric space, even if spatial information does not seem relevant to the task.

Regarding the neural correlates of the interaction between spatial and linguistic information, we suggest that the spatial manipulation of phonological material results in the activation of specialised attentional resources in the left hemisphere. This aligns with the findings Turriziani and colleagues [10] reported. The authors documented that the left parietal cortex could be the neural correlate that underpins the bias in attention to and the mental representation of semantic information. Moreover, this suggestion aligns with the “hemispheric activation model” [14,15], proposing that the distribution of attention in a space is biased contralaterally to the more activated hemisphere. Therefore, we speculate that verbal processing activates the left language-dominant hemisphere more strongly than it does the right hemisphere. This activation could be responsible for shifting attention towards the right hemispace. In line with this hypothesis, neuroimaging investigations have implicated a network involving the left hemisphere’s parietal and frontal areas in attention orientation in language tasks [16]. Again, this hypothesis aligns with the two lesion studies’ clinical reports. Indeed, the aphasic patient described by Coslett et al. [11] and the agrammatic patient described by Chatterjee et al. [12] had left parietal lesions.

The combination between visuospatial attention and semantic components has also been recently investigated in a TMS study [17], suggesting the left intraparietal sulcus as the neural correlate of such an interaction.

The recent report on the modulation of linguistic functions following left hemispheric activation via spatial adaptation procedures [18] also aligns with the model of interaction between spatial attention and language in the left hemisphere.

An interaction between spatial and linguistic information could also be interpreted under the theory of magnitude (ATOM) framework [19], proposing an interaction between different magnitudes (i.e., space, time, and numbers) both at a cognitive and neural level. Although linguistic material cannot be strictly interpreted as a magnitude, some models suggest that semantic information can be represented in vectorial terms [10,20]. According to this view, when a task requires manipulation of linguistic material in terms of semantic distance, spatial factors could interact with semantics or phonology in the same way they interact with numerical or time dimensions [21].

In conclusion, these findings provide valuable insights into the mapping of semantic and phonological information onto spatial representations within language processing. They emphasize the significance of considering spatial attention as a factor influencing language-related tasks.

Further research is warranted to delve deeper into the underlying mechanisms responsible for these attentional biases and to explore their implications for individuals with neurological or cognitive impairments. By unravelling the intricacies of the interaction between spatial attention and language processing, future studies can enhance our understanding of cognitive processes and potentially contribute to the development of interventions or therapies for individuals with language-related disorders.

## 5. Conclusions

This study demonstrated that spatial attentional biases could be observed during the processing of linguistic material, specifically in tasks involving the evaluation of phonological distance between words. The findings revealed that healthy subjects overestimate the phonological distance between the middle and outer words in the right space. Importantly, this bias was specific to word stimuli and was absent when pseudowords and consonant strings were used.

These results further support the notion that semantic and phonological information in language processing is internally mapped onto spatial representations. The observed attentional biases suggest that individuals allocate attention differently based on the spatial location of linguistic stimuli.

These findings contribute to understanding the complex relationship between language processing and spatial cognition. Overall, this study enhances our knowledge of how spatial representations play a role in processing linguistic material and highlights the importance of considering spatial attention in language-related tasks.

Further research is necessary to explore the underlying mechanisms responsible for these attentional biases and to investigate their implications for individuals with neurological or cognitive impairments.

## Figures and Tables

**Figure 1 brainsci-13-01123-f001:**
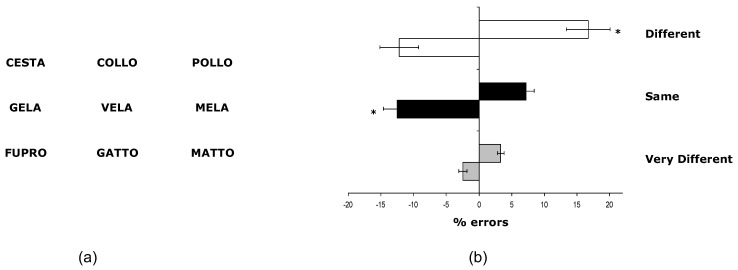
**Experiment 1.** (**a**) Example stimuli of Different condition (**top**), Same condition (**middle**), and Very Different condition (**bottom**) in the phonological distance judgment task; (**b**) mean leftward and rightward errors (±SE) as a function of the different experimental conditions. Negative values indicate leftward shifts, and positive values rightward shifts in judgment. The asterisk indicates the significance level (* *p* < 0.05).

**Figure 2 brainsci-13-01123-f002:**
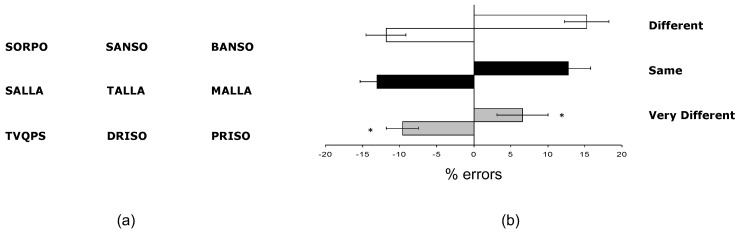
**Experiment 2.** (**a**) Example stimuli of Different condition (**top**), Same condition (**middle**), and Very Different condition (**bottom**) in the phonological distance judgment task; (**b**) mean leftward and rightward errors (±SE) as a function of the different experimental conditions. Negative values indicate leftward shifts, and positive values rightward shifts in judgment. The asterisk indicates the significance level (* *p* < 0.05).

**Table 1 brainsci-13-01123-t001:** The average number of “left”, “right” and “same” responses of participants in the different experimental conditions.

Condition		Responses	
	**same**	**left**	**right**
**Same (L = R)**	32.75 (2.76)	4.56 (2.52)	2.68 (1.95)
**Different (L < R)**	1.87 (2.36)	35.62 (3.94)	2.5 (2.82)
**Different (R < L)**	3 (2.55)	2.93 (2.67)	33.87 (4.52)
**Very Different (L < R)**	0	38.63 (1.58)	1.37 (1.58)
**Very Different (R < L)**	0	2.06 (1.65)	37.94 (1.65)

## Data Availability

Data are available upon request from the corresponding author.

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
