# Peer review of "Overestimation of Phonological Judgments on the Right Side of Space"

_brainsci, 2023, doi:10.3390/brainsci13081123_

Round 1

Reviewer 1 Report

Authors have proposed a manuscript titled “Overestimation of phonological judgments on the right side of 2 space”. The following comments should be incorporated into the manuscript.

1.    Proposed work should be incorporated in the introduction section.

2.    Recent literature should be added to the manuscript.

3.    FIg. 1 (a &b) should be explained in detail.

4.    I have seen two Experimental studies in the manuscript. Can you elaborate on the data in detail?

5.    Novelty is not clear in the manuscript. It should be explained in the manuscript.

6.    Comparative analysis can be done in tabular form.

7.    Conclusion should be made as per the result analyzed.

Author Response

First, we thank the reviewers for their comments that helped us improve the manuscript. Here we want to discuss how we have dealt with the points raised by the two reviewers.

Proposed work should be incorporated in the introduction section.

After a brief review of the literature on spatial attentional biases and their interactions with linguistic domains, the introduction section introduces a last paragraph stating the study's objective to investigate whether spatial biases could also affect phonological judgments. A brief introduction of the experimental paradigm used for addressing this topic is also presented.

  • Recent literature should be added to the manuscript.

Some references to recent work investigating interactions of spatial with linguistic domains in either healthy subjects or neurological patients are present in the discussion section. Please note that the literature on this topic is not so rich.

  • FIg. 1 (a &b) should be explained in detail.

We have added a detailed explanation of the results in the discussion section.

  • I have seen two Experimental studies in the manuscript. Can you elaborate on the data in detail?

We have tried to explain the data better, thanks to a table showing the number of responses the subjects gave in the different experimental conditions.

  • Novelty is not clear in the manuscript. It should be explained in the manuscript.

In the Discussion section, after summarizing the main results, we have added a sentence stating what is, in our opinion, the main novelty of the study: to provide the first demonstration that phonology is a language component that can be processed concerning spatial components.

  • Comparative analysis can be done in tabular form.

Please see point 4.

  • The conclusion should be made as per the result analyzed.

A conclusive summary of the main results is presented in the results section at the end of each paragraph, showing the results of each experiment. The conclusion at the end of the Discussion section presents general conclusions about the implications of these findings for a better understanding of the complex relationships between language processing and spatial cognition.

We thank the reviewer since these observations helped us to try to improve the clarity of the manuscript.

In each experiment, each subject faces three conditions: 1. triplets with the Same phonological distance between the middle and either outer word (Same Condition); 2. Triplets with a phonological distance shorter between the middle and the left outer word (Different Condition); 3. Triplets with a phonological distance shorter between the middle and the right outer word (Different Condition).

The Very Different Condition is a Different Condition in which one outer stimulus is a pronounceable pseudoword. In this sense, the very Different condition is a control condition with a much smaller number of errors.

For each kind of stimulus, a subject gives one response on each trial, as correctly observed by the reviewer. The responses could be of three types: Same; Left (i.e. shorter to the left); Right (i.e. shorter to the right).

In each trial, this response can be correct or wrong. In the case of a wrong response, the error could imply an overestimation of phonological distance in the right or left space.

Let’s consider the Same condition to take the example of the reviewer. In the Same condition, the two outer words have the same sound as the middle one. Therefore, the only correct response is the response “Same”. The response “Left” is an error that considers shorter the phonological distance between the left outer word and the middle word, i.e. it overestimates the phonological distance in the right hemifield (or it underestimates it in the left hemifield). The response “Right” is an error that considers as shorter the phonological distance between the right outer word and the middle word, i.e. it overestimates the phonological distance in the left hemifield (or it underestimates it in the right hemifield).

Let’s now consider the Different (or Very Different) conditions. In these conditions, again, the subject has three response options: same, left, and right. The “same” response is always an error: when the phonological distance is, in fact, shorter between the middle word and the right outer word, a response “same” is an error that overestimates the phonological distance in the right hemifield (or it underestimates it the left hemifield); when the phonological distance is, in fact, shorter between the middle word and the left outer word, a response same is an error that overestimated the phonological distance in the left hemifield (or it underestimates it in the right hemifield). The “left” response is an error only when the phonological distance is shorter between the middle and the right outer word. The “right” response is an error only when the phonological distance is shorter between the middle and the left outer word.

To facilitate comprehension of the results and analysis, as requested by both reviewers, in addition to the previous examples added to the text, we have added a Table reporting the average responses of the subjects (same, left or right) in each experimental condition.

We hope these revisions significantly improved the manuscript, making it acceptable for publication in your Journal.

With best regards

Reviewer 2 Report

Review of ‘Overestimation of phonological judgments on the right side of space

Patrizia Turriziani, Alessia Santostefano, Angela Catania and Massimiliano Oliveri.

These authors had previously reported that subjects overestimate the semantic distance between word stimuli in the right versus left hemifield; here they followed up by comparing  phonological distance between words in the right and left hemifield. Real words, pseudowords, and nonsense strings were tested; only real words showed an effect.  The study was well done, the English language is clear, the number of subjects is adequate, and the outcome is potentially interesting and surely novel. I will recommend publication, once I can be made to understand the presentation of the results (Fig. 2).

 Concerns (some minor).

Abstract: uses ‘space’ and ‘hemispace’, but you mean ‘hemifield’, given that fixation was always central.

Line 45: since it is as easy to attend to a stimulus X deg to the left of the fovea as to the same stimulus presented X deg to the right of the fovea, the term ‘attention’ might better be replaced by the more general term, ‘processing’ (although if the authors of the cited study state ‘attention’, then that is what must be quoted).

Line 47-50: “Healthy subjects were  presented with three pictures of items in the same semantic category arranged horizontally (one middle and two outer pictures). They were asked to indicate the spatial position  in which the semantic distance between the outer and middle pictures was smaller.

The results showed an overestimation of the semantic distance of items on the right side of pace.”

   The grammar in the second sentence need fixing. May I suggest (but please check I got it right)::

Healthy subjects viewed three pictures of items in the same semantic category arranged horizontally, one on the left side, one in the middle, and one on the right side. On average, the semantic distance between the middle picture and left one was reported as smaller than the semantic distance between the middle picture and the right one.”

Re line 52:  note that this effect does NOT imply that one side was more attended than the other. (Actually I doubt that emphasizing one side would affect judged semantic distance, but this is a matter to test.)

Re line 57: I can see that hemispheric specialization could influence perceived semantic and phonological distances (presumably in opposite directions), but again, I don’t see how this can relate to ‘attention’. In all cases, subjects attend to the stimuli; they just judge them differently depending on spatial position.

Line 74: I presume the monitor ran at 50 Hz ?  if so, this equals 11 frames.

Line 75: Ambiguous.  Specify if the central letters of the outer words were 5 deg eccentric, or the gap from the fovea to the first letter of each outer word was 5 deg, or the gap between the middle and either outer word was 5 deg.

Ine 82. By ‘phonological distance’, you mean the number of identical letters; the fewer, the greater the distance.  I think this needs explicit definition, because to me, the ‘distance’ refer to the sound difference between any pair of phonemes, e.g. “the distance between b and v is smaller than the distance between b and x. “   In the example mela, vela, gela, I would think that m/v was closer than v/g (although this may not be true in Italian), but here, they are treated as identical.

 English is, of course, a dreadful language to pronounce: consider that ‘ate’ and ‘eight’ sound the same (i.e., same phonemes, different letters). I don’t know how often this happens in Italian –maybe ignorable.

Line 90: I presume the pseudo words were all pronounceable ?

Line 95. (“Where is the word phonologically closest to the middle word?”).  Please explain the training procedure. Most students do not know the term ‘phonologically closest’ unless they have studied linguistics, and you do not say that they have.

Line 104. ’ We performed a 3x2 ANOVA on the mean number of errors, with the variables Condition (Different, Same, Very Different) and Space (left, right) as within-subjects factors.’

  I do not understand the relationship between this analysis and Fig. 2.   The problem for me is that the subject only gives one response on each trial, namely that either the ‘right’ or ‘left’ distance is the greater, and on every trial, the subject sees all 3 stimuli (left, middle, and right).

 Consider ‘mela, vela, gela’. The subject should respond ‘left’. If the subject responds ‘right’, was it because he mis-coded ‘mela, vela’ as a large distance, or miscoded ‘vela, gela’ as a small one ? So, where was the error – left or right ?  Impossible to tell.

In fig. 2, the ‘same’ condition, errors are possible, because subjects are also given the option of reporting ‘same’, so any report of ‘different’ is an error. However, as the phonological distance on the left equals that on the right, the assignment of error to side again seems impossible, there being no ‘right answer’ .  Yet fig. 2 shows more ‘same’ errors on the left than on the right. I don’t see how.

   Whatever the answer, including perhaps that I am being rather stupid, I still think that the definition of ‘error’, and exactly how they are counted, should be clarified.

  Perhaps a tabulation would help, of the numbers of ‘same’, ‘left’, and ‘right’ responses in 3 columns, the 3 rows indicating the stimulus (i.e., L>R, L<R, L=R), where L=left and R=right. Given such a table, how the proportions were obtained for the ANOVA, and for Fig. 2, could be made explicit.

Remark: In principle, some errors will be ‘false alarms’, meaning that a zero distance is taken to be large; and some will be ‘misses’, meaning that a large distance is ignored and the subject thinks that there is no difference. The term ‘error’ used here encompasses both possibilities, but separating them out in future research might help establish whether sensitivity (d’) or bias (beta or c) is differs from left to right. 

Author Response

First, we thank the reviewers for their comments that helped us improve the manuscript. Here we want to discuss how we have dealt with the points raised by the two reviewers.

Abstract: uses ‘space’ and ‘hemispace’, but you mean ‘hemifield’, given that fixation was always central.

We agree with the reviewer, and we have modified the abstract accordingly.

Line 45: since it is as easy to attend to a stimulus X deg to the left of the fovea as to the same stimulus presented X deg to the right of the fovea, the term ‘attention’ might better be replaced by the more general term, ‘processing’ (although if the authors of the cited study state ‘attention’, then that is what must be quoted).

Even in this case, we agree with the reviewer and have modified this section of the introduction accordingly. On the other hand, we continue to refer to attentional processing concerning what is reported in the cited study and also according to what we consider the general implications of this study, i.e. that attentional processes could mediate the complex relationships between language processing and spatial cognition.

Line 47-50: “Healthy subjects were presented with … The grammar in the second sentence need fixing.

We have accepted the option presented by the reviewer and modified the text accordingly.

Re line 52:  note that this effect does NOT imply that one side was more attended than the other. (Actually I doubt that emphasizing one side would affect judged semantic distance, but this is a matter to test.)

This is an interesting point to be tested in future studies, and we thank the reviewer for pointing it out.

Re line 57: I can see that hemispheric specialization could influence perceived semantic and phonological distances (presumably in opposite directions), but again, I don’t see how this can relate to ‘attention’. In all cases, subjects attend to the stimuli; they just judge them differently depending on spatial position.

We suggest that hemispheric specialization influences perceived semantic and phonological distances in the same direction. We agree that perhaps the reference to attention for explaining these interactions could be too speculative at this stage. We have removed the term attention from this paragraph, as suggested.

Line 74: I presume the monitor ran at 50 Hz ? if so, this equals 11 frames.

The reviewer’s observation is correct; we have specified this detail in the text.

Line 75: Ambiguous. Specify if the central letters of the outer words were 5 deg eccentric, or the gap from the fovea to the first letter of each outer word was 5 deg, or the gap between the middle and either outer word was 5 deg.

We have specified that the gap between the middle and either outer word was 5 deg.

Line 82. By ‘phonological distance’, you mean the number of identical letters; the fewer, the greater the distance. I think this needs explicit definition because to me, the ‘distance’ refer to the sound difference between any pair of phonemes, e.g. “the distance between b and v is smaller than the distance between b and x. “   In the example mela, vela, gela, I would think that m/v was closer than v/g (although this may not be true in Italian), but here, they are treated as identical.

In fact, by phonological distance we refer to the difference in the “sound” of phonemes of the whole word. In the Italian language, where there is quite a close correspondence between graphemic and phonemic representations in terms of sound, differently from what happens in English, this could correspond in some cases to the number of identical letters (as it happens in the “Same” condition). However, the explicit instruction and experimental training given to the subjects before entering the main experiment referred to phonological distance regarding sound differences between the whole word. We have tried to better specify this crucial point in the Methods section.

Line 90: I presume the pseudo words were all pronounceable ?

The reviewer’s observation is correct; we have specified this detail in the text.

Line 95. (“Where is the word phonologically closest to the middle word?”). Please explain the training procedure. Most students do not know the term ‘phonologically closest’ unless they have studied linguistics, and you do not say that they have.

See the previous point about the meaning of phonological distance in these experiments and how it was explained to the experimental subjects.

I do not understand the relationship between this analysis and Fig. 2.   The problem for me is that the subject only gives one response on each trial, namely that either the ‘right’ or ‘left’ distance is the greater, and on every trial, the subject sees all 3 stimuli (left, middle, and right).

 Consider ‘mela, vela, gela’. The subject should respond ‘left’. If the subject responds ‘right’, was it because he mis-coded ‘mela, vela’ as a large distance, or miscoded ‘vela, gela’ as a small one ? So, where was the error – left or right ? Impossible to tell.

We thank the reviewer since these observations helped us to try to improve the clarity of the manuscript.

In each experiment, each subject faces three conditions: 1. triplets with the Same phonological distance between the middle and either outer word (Same Condition); 2. Triplets with a phonological distance shorter between the middle and the left outer word (Different Condition); 3. Triplets with a phonological distance shorter between the middle and the right outer word (Different Condition).

The Very Different Condition is a Different Condition in which one outer stimulus is a pronounceable pseudoword. In this sense, the very Different condition is a control condition with a much smaller number of errors.

A subject gives one response on each trial for each kind of stimulus, as correctly observed by the reviewer. The responses could be of three types: Same; Left (i.e. shorter to the left), and Right (i.e. shorter to the right).

In each trial, this response can be correct or wrong. In the case of a wrong response, the error could imply an overestimation of phonological distance in the right or left space.

Let’s consider the Same condition, to take the example of the reviewer. In the Same condition, the two outer words have the same sound as the middle one. Therefore, the only correct response is the response “Same”. The response “Left” is an error that considers shorter the phonological distance between the left outer word and the middle word, i.e. it overestimates the phonological distance in the right hemifield (or it underestimates it in the left hemifield). The response “Right” is an error that considers as shorter the phonological distance between the right outer word and the middle word, i.e. it overestimates the phonological distance in the left hemifield (or it underestimates it in the right hemifield).

Let’s now consider the Different (or Very Different) conditions. In these conditions, the subject has three response options: same, left, and right. The “same” response is always an error: when the phonological distance is, in fact, shorter between the middle word and the right outer word, a response “same” is an error that overestimates the phonological distance in the right hemifield (or it underestimates it the left hemifield); when the phonological distance is, in fact, shorter between the middle word and the left outer word, a response same is an error that overestimated the phonological distance in the left hemifield (or it underestimates it in the right hemifield). The “left” response is an error only when the phonological distance is shorter between the middle and the right outer word. The “right” response is an error only when the phonological distance is shorter between the middle and the left outer word.

To facilitate comprehension of the results and analysis, as requested by both reviewers, in addition to the previous examples added to the text, we have added a Table reporting the average responses of the subjects (same, left or right) in each experimental condition.

We hope these revisions significantly improved the manuscript, making it acceptable for publication in your Journal.

With best regards

Round 2

Reviewer 1 Report

Authors are incorporated suggestions to the manuscript.